

# Global nitrogen and phosphorus fertilizer use for agriculture production in the past half century: Shifted hot spots and nutrient imbalance
**Chaoqun Lu[1, 2*] and Hanqin Tian[2]**
[1]Department of Ecology, Evolution, and Organismal Biology, Iowa State University, Ames, IA
50011, USA;
[2]International Center for Climate and Global Change Research, and School of Forestry and
Wildlife Sciences, Auburn University, Auburn, AL, 36849, USA;
*Corresponding author, email: clu@iastate.edu



## Abstract

In addition to enhance agricultural productivity, synthetic nitrogen (N) and phosphorous

(P) fertilizer application in croplands dramatically altered global nutrient budget, water quality,

greenhouse gas balance, and their feedbacks to the climate system. However, due to the lack of

geospatial fertilizer input data, current Earth system/land surface modeling studies have to ignore

or use over-simplified data (e.g., static, spatially uniform fertilizer use) to characterize

agricultural N and P input over decadal or century-long period. In this study, we therefore

develop a global time-series gridded data of annual synthetic N and P fertilizer use rate in

croplands, matched with HYDE 3,2 historical land use maps, at a resolution of 0.5° latitude by

longitude during 1900-2013. Our data indicate N and P fertilizer use rates increased by

approximately 8 times and 3 times, respectively, since the year 1961, when IFA (International

Fertilizer Industry Association) and FAO (Food and Agricultural Organization) survey of

country-level fertilizer input were available. Considering cropland expansion, increase of total

fertilizer consumption amount is even larger. Hotspots of agricultural N fertilizer use shifted

from the U.S. and Western Europe in the 1960s to East Asia in the early 21st century. P fertilizer

input show the similar pattern with additional hotspot in Brazil. We find a global increase of

fertilizer N/P ratio by 0.8 g N/g P per decade ($p < 0.05$) during 1961-2013, which may have

important global implication of human impacts on agroecosystem functions in the long run. Our

data can serve as one of critical input drivers for regional and global assessment on agricultural

productivity, crop yield, agriculture-derived greenhouse gas balance, global nutrient budget,

land-to-aquatic nutrient loss, and ecosystem feedback to the climate system. Datasets available

at: https://doi.pangaea.de/10.1594/PANGAEA.863323





## Introduction


Agricultural fertilizer use is one of important land management practices that alleviated
nitrogen limitation in cropland and substantially increased crop yield and soil fertility over the
past century (Vitousek et al., 1997; Tilman et al., 2002). Since the generation of Harber-Bosch
process in the early 20$^{th}$ century, chemical nitrogen (N) fertilizer production has converted large
amount of unreactive N to reactive forms (Galloway et al., 1997). Chemical phosphorus (P)
fertilizer production was promoted as well with the phosphorus acid. On one hand, as critical
component of "Green Revolution", the dramatic increase in fertilizer production and application
has contributed considerably to raise agricultural productivity and reduce hunger worldwide
(Smil, 2002; Erisman et al., 2009). On the other hand, excessive fertilizer use is proven to cause
a number of environmental and ecological problems within and outside of farmlands, such as air
pollution, soil acidification and degradation, water eutrophication, crop yield reduction, and
undermine the sustainability of food and energy production from the field (Ju et al., 2009;
Vitousek et al., 2009; Guo et al., 2010; Sutton et al., 2011; Tian et al., 2012; Lu and Tian, 2013)
Large spatial and temporal variations exist in chemical fertilizer use across the world.
China, United States, and India together accounted for over 50% of fertilizer consumption
globally and they demonstrated contrasting changing trend over the past century due to the status
of economic and agricultural development (FAOSTAT, 2015). The rates and spatiotemporal
patterns of N and P fertilizer uses are one of key input drivers for inventory- and process-based
land modeling study to reliably estimate agroecosystem processes (Mosier et al., 1998; Zaehle et
al., 2011; Stocker et al., 2013; Tian et al., 2015). N input-related processes affect a wide variety
of plant physiological, biogeochemical and hydrological variables (e.g., crop productivity, yield,
evapotranspiration, N$_2$O emission, N and P leaching from agricultural runoff and land-to-aquatic





export of N and P) and their responses to other environmental drivers (e.g., $CO_2$ fertilization
effect). However, there is still a lack of dataset to describe long-term spatially-explicit
agricultural input of N and P through chemical fertilizer use across the globe.

IFA and FAO provide data of annual fertilizer consumption amount across croplands

since 1961, which is the most complete country-level record of fertilizer use over a long time
period. By assuming uniform fertilizer application rate nationwide, multiple process-based
modeling studies considering management practices (Zaehle et al., 2011; Stocker et al., 2013)
have used this data set as an important driver for agroecosystems, however, the spatial variations
in fertilizer use within countries have been overlooked. Tian et al. (2015) has updated FAO-
based fertilizer use data by using detailed regional information in China, India and USA to
replace country-uniform data and keeping the rest countries the same as FAO statistics. They
partially demonstrated within-country variations through province-level census in China, and
state-level census in India and U.S. (Tian et al., 2011; Lu and Tian, 2013; Banger et al., 2015).
Based on country-level crop-specific fertilizer record (''Fertilizer Use by Crop 2002'', from
IFADATA) and global distribution map of 175 crops (Monfreda et al., 2008), Potter et al. (2010)
generated annual N and P fertilizer application data across the globe at a spatial resolution of 0.5°
in latitude by longitude. This data contains most of crop-specific variations in N and P fertilizer
use over space, but it only represents average fertilizer application pattern in the period of 1994
to 2001 and couldn't meet the time frame of long-term land surface modeling. Likewise, Muller
et al. (2012) used similar approach to distribute crop- and crop group-specific fertilizer use rate,
and combine multi-source national and sub-national nutrient consumption data to harmonize
fertilizer use rate. However, their data only represent the status around 2000. Therefore, in this
study, we develop a spatially-explicit time-series N and P fertilizer use data by combining the





country-level fertilizer use record, crop-specific fertilizer use data, global maps of annual
cropland area, and spatial distribution of crop types at a 0.5º-resolution during the period 1900-
2013. This newly-developed data set displayed within-country heterogeneity of fertilizer use
while keeping the country-level total fertilizer consumption amount consistent with IFA data,
and it has been recently incorporated as one of key environmental drivers for global model
simulation studies and model-model intercomparison project (e.g., $N_2O$-MIP, Tian et al. *in prep*).
To facilitate Earth System Modeling and inventory-based studies, this global N fertilizer use data
will be updated annually based on the most recent IFA/FAO country-level statistics data and
historical land use maps.
**Methods**

The basic principle is to spatialize the country-level N and P fertilizer use amount to

gridded maps of fertilizer use rate on per unit area cropland during the period 1961-2013 (Figure
1), in which IFA and FAO have annual record for most countries. Here we adopt "Grand Total N
and $P_2O_5$" from IFA statistics data in the unit of thousand tonnes nutrients for each country. The
"Grand total" amount includes nutrients from straight and compound forms. N fertilizer use rate
before 1910 is set to be 0, and the data between 1911 and 1961 is assumed to linearly increase in
each pixel. P fertilizer use rate is assumed to linearly increase between 1900 and 1961. We
convert g $P_2O_5$ in IFA database and Heffer (2013) to g P by multiplying the ratio of 62/142.

**Crop-specific N and P fertilizer use rate:** The database of crop-specific N and P fertilizer

use from IFA (Heffer, 2013) provides the total amount of N fertilizer use in 13 crop groups at
country level, which includes 27 selected countries (considering EU-27 as a single countries,
Figure 2) in the year of 2010-2010/11. It accounts for over 94% of global fertilizer consumption.
M3-crops data developed by Monfreda et al. (2008) depicts harvest area of 175 crops in the year





of 2000 at 5-arc min resolution in latitude by longitude. Its unit is proportion of grid cell area and
the values could be larger than 1 because of multiple cropping. We calculated the harvested area
of these 13 crop groups (i.e., wheat, rice, maize, other cereal, soybean, oil palm, other soil seed,
fiber, sugar, roots, fruit, vegetable, and others) in the corresponding 26 countries and EU-27. We
obtained country-level crop-specific N and P fertilizer use rate, by dividing crop-specific
fertilizer consumption amount by harvested area of each crop group. Here, by using harvested
area, instead of area of arable land, we consider the effect of multiple cropping on the calculation
of N fertilizer use rate to avoid overestimating N input in cropland. This tabular data was
interpolated to generate spatial maps of N and P fertilizer use rate for each crop group.
Combining with harvested area of each crop, we produced the area-weighted average of N and P
fertilizer use rate in each grid cell, which will serve as a baseline map to downscale country-level
fertilizer use.
$$\overline{C_{Nfer}}_g = \frac{\sum_i (\frac{C_{Nfer_{i,j}}}{A_{harv_{i,j}}} \times A_{harv_{i,g}})}{\sum_i A_{harv_{i,g}}}$$

Where $\overline{C_{Nfer}}_g$ is average crop-specific nutrient (N and P) fertilizer use rate (g N or g
P/m$^2$/yr) at grid level, $C_{Nfer}$ and $A_{harv}$ are crop-specific N and P fertilizer use amount (g N or g
P) and harvested area (m$^2$), respectively, for crop type $i$, country $j$, and grid cell $g$ (Figure 1).
**IFA-based national fertilizer use interpolation:** We divided country- and continent-scale
annual fertilizer consumption amount from IFA by annual cropland area calculated from HYDE
3.2 (Klein Gildewijk, 2016) to get half-degree gridded N and P fertilizer use rate during 1961-
2013. In this step, we assume the N and P fertilizer is evenly applied in croplands of each
country. To represent the status of countries not included in IFA, the amount of fertilizer





application in IFA-included countries was subtracted from continental total, and the rest fertilizer
was assumed to be evenly applied in croplands not covered by IFA country-level survey. These
non-IFA countries together cover ~8% of global croplands, and account for less than 1% of
global synthetic N and P fertilizer consumption. Several countries (e.g., former Soviet Union,
former Czechoslovakia, former Yugoslavia) was broken up in the 1990s, and the emergent
countries only have fertilizer use archived thereafter. We use average fertilizer use rate at per
unit cropland area in the former countries to represent new countries' agricultural nutrient input
before their existence.
**Harmonizing national total and crop-specific fertilizer use rate:** In order to keep the
national total N and P fertilizer amount consistent with IFA inventory, we calculated country-
level ratios between the time-series (1961-2013) national fertilizer use amount from IFA and the
product of gridded fertilizer use rate ($\overline{C_{Nfer_g}}$) and gridded cropland area delineated by HYDE
3.2. This tabular country-level regulation ratio data was interpolated to half-degree maps,
combined with gridded fertilizer use rate ($\overline{C_{Nfer_g}}$), for generating spatially-variant N and P
fertilizer use rate during 1961-2013. This approach was only used in the grid cells containing
croplands according to HYDE 3.2. In the rest areas, fertilizer use rate is zero.
$$R_{Nfer_{y,j}} = \frac{CTY_{Nfer_{y,j}}}{\sum_{g=1}^{g=n\ in\ country\ j}(\overline{C_{Nfer_g}} \times A_{crop_{y,g}})}$$

Where $R_{Nfer_{y,j}}$ is the regulation ratio (unitless) in the year $y$ and country $j$. $CTY_{Nfer_{y,j}}$ is
national total N fertilizer use amount (unit: g N/yr or g P/yr) derived from IFA database in a
specific year, and $A_{crop_{y,g}}$ is the area of cropland (unit: m$^2$) retrieved from the historical half-
degree land use data (HYDE 3.2) in the year of $y$ and grid of $g$.



$$Nfer_{y,g} = \overline{C_{Nfer}} \times R_{Nfer_{y,g}}$$
Where gridded N and P fertilizer use rate (unit: g N or P/m$^2$ cropland/yr) in the year $y$ and
grid $g$ is the product of average crop-derived N fertilizer use rate and the modification ratio
$(R_{Nfer_{y,j}})$ in corresponding year and grid cell.
It is notable that EU-27 has the same crop-specific fertilizer use rate for each crop group,
but IFA-based country-level fertilizer use amount is different among countries and years, and
thus annual maps of regulation ratios are different spatially. Therefore, the final product shows
spatially variant N and P fertilizer use rate in the region of EU-27.
**Results**
Our data indicates that N fertilizer consumption increased from 11.3 Tg N/yr (0.9 g N/m$^2$
cropland/yr) in 1961 to 107.6 Tg N/yr (7.4 g N/m$^2$ cropland/yr on average) in 2013, and that P
fertilizer consumption increased from 4.6 to 17.5 Tg P/yr (0.4 to 1.2 g P/m$^2$ cropland/yr on
average) during the same period (Figure 3). Increase of global total fertilizer use amount is
derived from both cropland expansion and raised fertilizer application rate in per unit cropland
area. In 2013, the top five fertilizer-consuming countries (China, India, U.S., Brazil, and Pakistan
for N fertilizer, and China, India, U.S., Brazil, and Canada for P fertilizer) together accounted for
63% of global fertilizer consumption. China alone shared 31% of global N fertilizer consumption
with an annual increasing rate of 0.7 Tg N/y or 0.6 g N/m$^2$ cropland/yr ($R^2 = 0.98$) during 1961-
2013 (Figure 4), while India showed a much smaller increasing trend of 0.3 Tg N/yr or 0.2 g
N/m$^2$ cropland/yr per year ($R^2 = 0.97$). N fertilizer use rate in the U.S. increased by 0.4 Tg N/yr
or 0.2 g N/m$^2$ cropland/yr per year during 1961-1980 and leveled off thereafter. P fertilizer use in
these three countries demonstrated similar pattern: more rapid increase in China (0.1 Tg P/yr)



than that in India (0.06 Tg P/yr) and the U.S (0.05 Tg P/yr during 1961-1980 and leveled off
thereafter). Brazil accounted for 3% and 11% of global N and P fertilizer consumption,
respectively. N fertilizer use rate in Brazil gradually increased since the early 1990s, and now
reached half of the agricultural N input level in the U.S., while its P fertilizer use rate ranked the
global top in 1980, declined thereafter, and regrew from 2000, demonstrating the second highest
per unit cropland P fertilizer use rate next to China. Pakistan shared 3% of global total N
fertilizer use, but its average cropland application rate increased dramatically with an annual
increase rate of 0.3 g N/ $m^2$ cropland/yr ($R^2 = 0.97$), only next to China (Figure 4).

Agricultural N fertilizer use rate was peaked in the U.S. and western Europe in the 1960s,

and the hot spots gradually moved to Western Europe and East Asia in the 80s and 90s, and then
to East Asia in the early 21$^{st}$ century (Figure 5). Large area of croplands in East and Southeast
China stands out due to extremely high N fertilizer input (e.g., more than 30 g N/$m^2$/yr). The
northern India and western Europe received 10-20 g N/$m^2$/yr up to now. South America also
experienced rapid increase of N fertilizer use rate during the past 54 years, particularly for small
areas of Brazil, with N input reaching the similar level as the U.S. Although cropland expansion
widely occurred in Africa, its average N fertilizer use rate was enhanced slowly, with most areas
still receiving less than 1.5 g N/$m^2$/yr in 2013. Australia demonstrated the similar low level of
agricultural N input (less than 5 g N/$m^2$/yr in 2013). N fertilizer use in Russia peaked in the
1980s, and then declined in the following decades. It is argued that, after 1990, the major reason
for fertilizer use drop is a severe economic depression due to the breakup of Soviet Union and
the following conversion to market economies (Ivanova and Nosov, 2011).

Europe was hot spot of agricultural P fertilizer input before the 1980s, and it shifted to

Central China and small area of Brazil with input rate more than 3 g P/$m^2$ cropland/yr in 2013





(Figure 6). P input in China showed a significant increasing trend during 1961-2013 ($p < 0.05$),
while in Brazil, it peaked in the early 1980s and declined thereafter, and grew again since 2000.
Most agricultural areas across the rest of world were characterized by P input of less than 1 g
P/m² cropland /yr, except India, Western Europe, and small area of the U.S. receiving 1-1.5 g
P/m² cropland /yr in 2013. P fertilizer use rate remains relatively stable in the U.S. since 1980.
Similar to agricultural N fertilizer use, the increase of total P fertilizer amount in Africa was
primarily driven by cropland expansion, its input rate on per unit cropland area was constantly
low, less than 0.5 g P/m²/yr during the past half century. Likewise, P fertilizer use rate in Russia
increased in the 1980s, and began to decline after 1990.

We find the enhancement of N fertilizer use is faster than that of P fertilizer use, leading

to an increase of N/P ratio in synthetic fertilizer consumption from 2.4 to 6.2 g N/g P (an
increase of 0.8 g N/g P per decade, $p < 0.05$) during 1961-2013. This increase mainly took place
in Europe, North Asia, and small areas of South America and Africa (Figure 7). However,
fertilizer N/P ratio declined in China and India from over 9 g N/g P in 1961 to 5-9 g N/g P at
present, which is mainly caused by extremely low P fertilizer input in these two countries before
1980. It remained relatively stable in the U.S. and most countries of Africa since 1980. Up to
now, fertilizer N/P ratio in Northern Hemisphere is generally higher (more than 5) than that in
Southern Hemisphere.
**Discussion**

*Comparison with other studies:* In this study, we use M3-crop to spatialize crop-specific

fertilizer use rate and then use HYDE 3.2 to disaggregate the annual national IFA fertilizer use
record to grid cells with cropland. Therefore, the changes in fertilizer use rate shown in our data
could reflect the comprehensive human disturbances in cropland area and distribution, as well as



national total fertilizer inputs at annual time step (Figure 5 and 6). In addition, in spatializing
fertilizer data, the approach we used here based on crop-specific fertilizer use rate is more
reliable than national, provincial, state, or county-based fertilizer development which assumes
uniform fertilizer input rate in a certain region (Zaehle et al., 2011; Lu and Tian, 2013; Tian et
al., 2015). Regionally uniform rate has overlooked fertilizer use differences among crops. The 13
crop groups we adopted to spatialize national fertilizer use include the top fertilizer-consuming
crops (i.e., wheat, maize, soybean, rice, oil palm) and aggregate the rest of crops into other
cereal, other soil seed, fiber, sugar, roots, fruit, vegetable, and others, which keeps cross-country
cross-crop heterogeneity of fertilizer use in data development. Overall, combined with historical
land use data (e.g., HYDE 3.2), our century-long global maps at a $0.5º \times 0.5º$ resolution can be
used to force Earth System Models for assessing agroecosystem productivity, greenhouse gas
fluxes, N and P export through agricultural runoff, and their feedbacks to climate system.

This newly-developed database is based on IFA country-level time-series statistics and its

spatial distribution follows the pattern of crop-specific fertilizer use rate and gridded harvest area
of crop types in most of fertilizer-consuming countries. Our data are comparable to other existing
estimates in terms of N and P fertilizer consumption amount globally (Table 1). Our global total
is very close to IFA and FAO statistical data, and the slight differences in some years are derived
from mismatched cropland areas between FAO (Arable land and permanent crops) and HYDE
3.2.  Only a few existing data (e.g., Potter et al., 2010; Muller et al., 2012) characterize the
spatial heterogeneity and hot spots of N and P fertilizer use in agricultural land, but none of them
spans long enough to facilitate modeling study to capture the legacy effects of historical fertilizer
input. Potter et al. (2010) used the similar approach as we did and developed geospatial data of N
and P inputs from fertilizer and manure across the globe. But they didn't consider annual land



cover change and the resulting changes in spatial patterns of agricultural fertilizer use by using
one-phase M3-crop map which represents an average cropland distribution in the period 1997-
2003 (Monfreda et al., 2008). Likewise, Mueller et al. (2012) revised Potter's approach by
incorporating national and sub-national fertilizer application data for crops and crop groups,
harmonizing with FAO consumption record and allocating fertilizer to crop and pasture areas
derived from M3-crop map. Potter et al. (2010) and Mueller et al. (2012) both demonstrate total
N or P fertilizer use on per unit grid cell area, in order to compare them with our data in the year
of 2000, we converted these two data products to g of N or P on m$^{-2}$ of cropland area by dividing
grid-level total fertilizer amount by crop areas from M3-crop (Figure 8). We found the hot spots
of global N and P fertilizer use rate are roughly consistent among them. The major differences
are likely caused by the following reasons: 1) cropland area and distribution derived from HYDE
3.2 (used in our study) and M3-crop (used to delineate fertilizer use area in Potter et al., 2010
and Mueller et al., 2012) don't match in some areas, such as the western China, western U.S.,
Central Asia countries, North Africa, and Australia; 2) the crop-specific fertilizer use data in
2010-10/11 (Figure 2) used in our study covered more countries in North Asia, but less in Africa
and South America compared to IFA data from "Fertilizer Use by Crop 2002" in the
development of the other two data products, which led to different spatial details; 3) the IFA
crop-specific fertilizer use data in our study include 13 crop groups (i.e., major crops and groups
of "others") in each country (Figure 2), while crop types range from 2 to over 50 per country was
reported in the IFA crop fertilizer use data that is used in Potter et al. (2010) and Mueller et al.
(2012). Therefore, our data may to some extent diminish the cross-crop variations in fertilizer
application by using records of crop groups for these non-major crop types.



***Change in N and P fertilizer use:*** Global synthetic N and P fertilizer use increased by 85
Tg N/yr and 10 Tg P/yr, respectively, between the 1960s and recent 5 years (2009-2013). Across
the region, Southern Asia ( a region include East Asia, South Asia, and Southeast Asia, Figure 9)
accounted for 71% of the enhanced global N fertilizer use, followed by North America (11%),
Europe (7%), and South America (6%).  The other three continents shared the rest 5% increase.
Southern Asia is also the largest contributor (91%) to global P fertilizer use increase over the
past half century, followed by South America (21%) and North America (4%), while a decrease
in P fertilizer consumption (-17%) is found in Europe and neglible change in other continents.
Noticeably, Southern Asia ranks as a top hot spot of global anthropogenic nutrient input,
contributing to a number of ecological and environmental problems, such as increased
agricultural $N_2O$ emission, climate warming, nitrate and phosphate leaching, and coastal
eutrophication and hypoxia (Seitzinger et al., 2010; Bouwman et al., 2013; Tian et al., 2016).
N/P ratio in terrestrial plant species are 12-13 on average, with large cross-species and
cross-site variability (Elser et al., 2000; Knecht and Goransson, 2004). Human management,
such as fertilizer application can change N and P supply, and modify vegetation and soil
properties of N/P ratio and their responses to increased N input (Güsewell, 2004). Higher
fertilizer N/P in Northern Hemisphere (Figure 7) could be reasonably explained by faster N
fertilizer increase than P fertilizer in historically predominated N limitation and P-rich soil in
those areas. Particularly in Europe, P fertilizer use rate declined while N input continue
increasing. Fertilizer N/P ratio decline in China and India, however, indicates a shift from nearly
zero-synthetic P fertilizer input to gradually balanced fertilizer strategy (Zhang et al. 2005). In
contrast, South America is characterized by lower fertilizer N/P ratio because of its large
increase in both N and P fertilizer use (accounting f or 6% vs 21% of global increase since the

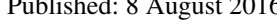
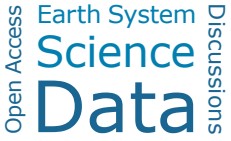

1980s, Figure 9). In the long run, global increase of anthropogenic N/P ratio is expected to
reduce species richness (Güsewell et al., 2005), induce the shift from N limitation to P limitation
(Elser et al., 2009; Peñuelas et al., 2012), and increase N loss (e.g., N loads to downstream
aquatic ecosystems, $NH_3$ volatilization and re-deposition elsewhere) due to the limitation of low
soil P availability to N fertilization effect (Carpenter et al., 1998). To better manage
agroecosystem productivity and its sustainability, the dynamic pattern of anthropogenic N/P
input ought to be related to local soil N and P status, growth demand od different crop species,
and historical nutrient inputs.
***Uncertainty and future needs:*** The uncertainties of this database are mainly from the
following aspects: (1) The data of country-level fertilizer use by crop we used in this study is the
latest estimate (i.e., 2010-2010/2011, Heffer, 2013), which could reflect current patterns of crop-
specific fertilizer application rate, but in the meanwhile may bias the historical allocation of
fertilizer use among crop groups. There is no long-term data indicating how variable the relative
contribution of crop groups is in consuming fertilizer at country level. Here, we assume that the
evolution of global crop production and crop area, rather than crop-specific fertilizer application
rate, is the major reason responsible for the share of fertilizer use among crops.  (2) The spatial
pattern of various crop types are derived from M3-crop (Monfreda et al., 2008), which is the
most complete and detailed distribution map of 175 crop types so far, though representing an
average status for 1997-2003. By using the information of distribution and harvested area for 13
crop groups from M3-crop, we convert crop-specific fertilizer use amount in each country to
gridded agricultural fertilizer use rate in per unit cropland area. The temporal mismatch between
fertilizer and crop distribution data may cause under- or overestimation of grid-level fertilizer
use rate. (3) We use HYDE 3.2 historical cropland percentage to allocate country-level fertilizer





use amount from IFA, but HYDE data is proven to show inconsistent spatial and temporal
patterns of cropland area change compared to satellite-derived land use database at regional scale
(e.g., China: Liu and Tian, 2010, and India: Tian et al., 2014, Figure 10). Based on high-
resolution satellite images and historical archives, the land use data from Liu and Tian (2010)
shows more concentrated cropland distribution with higher within-grid percentage in the
Northern China Plain, compared to HYDE 3.2, although national total cropland area is quite
similar between these two data in recent decade. This might be the reason that our data fail to
capture the extremely high fertilizer use rate in the Northern China Plain (more than 40 g N/m$^2$
cropland/yr as indicated in Lu and Tian, 2013 that used land use data from Liu and Tian, 2011).
In addition, the difference of national cropland area between HYDE3.2 and regional LCLUC
database (Figure 10) could make our fertilizer data underestimate average fertilizer use rate on
per m$^2$ cropland in India and overestimate fertilizer use rate before 1990 in China. As a result,
the extensive distribution of cropland and fertilizer use data in China derived from HYDE 3.2
may lead to uncertain estimates in Earth System Modeling. Therefore, we call for continuous
survey of crop-specific fertilizer use, development of dynamic crop type maps, and updated
global land use data with more precise regional description, for further improving
characterization of geospatial and temporal patterns of agricultural fertilizer use.
**Conclusion**
Synthetic N and P fertilizer application during agricultural production is a critical
component of anthropogenic nutrient input in the Earth system. Development of spatially-
explicit time-series N and P fertilizer uses across global cropland reveals a significant and
imbalanced increase of N and P during past half century (1961-2013). The nutrient input hot
spots shifted from North American and European countries to East Asia, which implies



corresponding changes in the spatial pattern of global nutrient budget, carbon sequestration and
storage, greenhouse gas emissions, and riverine nutrient export to downstream aquatic systems.
Meanwhile, Africa is still characterized by low nutrient input along with expanding cropland
areas. The increased fertilizer N/P ratio is likely to alter the nutrient limitation status in
agricultural land, and affect ecosystem responses to future N enrichment in the long run.
Agricultural management practices should put emphasis on increasing nutrient use efficiency in
those high input regions, while reducing environmental and ecological consequences of
excessive nutrient loads, and enhancing agricultural fertilizer application to relieve nutrient
limitation in low input regions. In addition to spatially balanced fertilizer use, balanced N:P:K
fertilizer application ought to be promoted depending on local nutrient availability and crop
growth demands.
**Acknowledgement**
This work is supported by Iowa State University new faculty start-up funds, NASA
Interdisciplinary Science Program (NNX10AU06G, NNX11AD47G, NNG04GM39C), NASA
Land Cover/Land Use Change Program (NNX08AL73G), NASA Carbon Monitoring System
Program (NNX14AO73G). We thank anonymous reviewers for their precious comments and
constructive suggestions to improve this manuscript.




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





Table 1 Comparison of synthetic N and P fertilizer use amount between this study and other
existing data sources.

| Data source | other estimates | This Study | Year |
|---|---|---|---|
| Synthetic N fertilizer amount (Tg N/yr) | | | |
| Van der Hoek and Bouwman, 1999 | 73.6 | 70.4 | 1994 |
| Sheldrick et al., 2002 | 78.2 | 80.3 | 1996 |
| Boyer et al., 2004 | 81.1 | | |
| Green et al., 2004 | 78.3 | 76.2 | 1995 |
| Siebert 2005 | 72.3 | | |
| Bouwman et al., 2005 | 82.9 | | |
| Potter et al., 2010 | 70.2 | | |
| Mueller et al., 2012 | 77.8 | 80.1 | 2000 |
| IFA | 82.1 | | |
| FAO stat | 80.8 | | |
| IFA | 110.2 | 107.6 | 2013 |
| FAO stat | 99.6 | | |
| Synthetic P fertilizer amount (Tg P/yr) | | | |
| Sheldrick et al., 2002 | 12.7 | 13.2 | 1996 |
| Smil, 2000 | 15 | | |
| Bouwman et al., 2009 | 13.8 | | |
| Potter et al., 2010 | 14.3 | 13.9 | 2000 |
| Mueller et al., 2012 | 13.7 | | |
| IFA | 14.3 | | |
| FAO stat | 14.2 | | |
| IFA | 18.8 | 17.5 | 2013 |
| FAO stat | 16.7 | | |





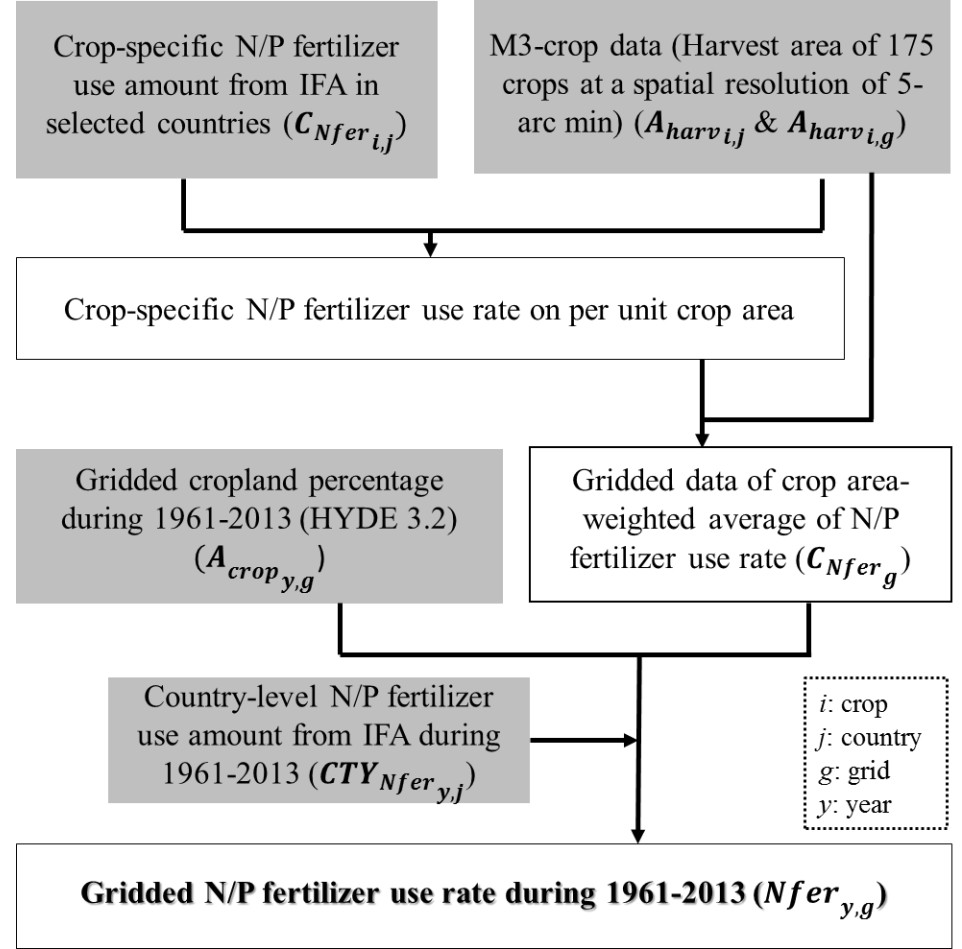


Figure 1 Diagram of the workflow for developing the global N fertilizer use rate data during the
period 1961-2013. The gray boxes indicate the raw data involved in N fertilizer data
development







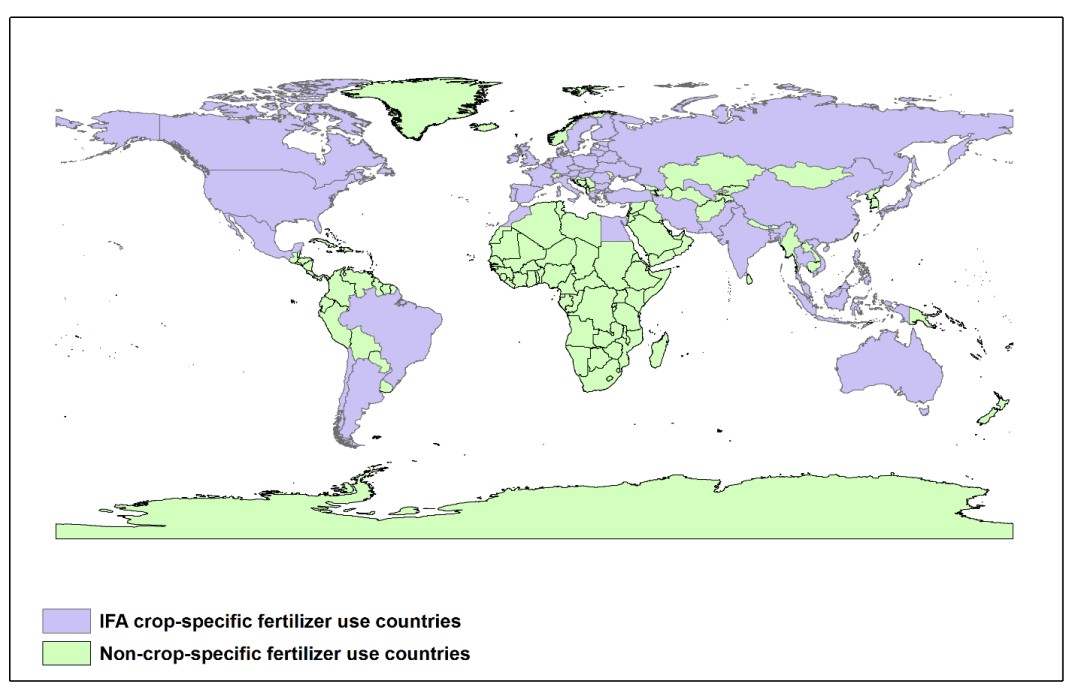


Figure 2 Countries with and without crop-specific fertilizer use records from IFA database in the
year 2010-10/11 (Heffer et al., 2013)






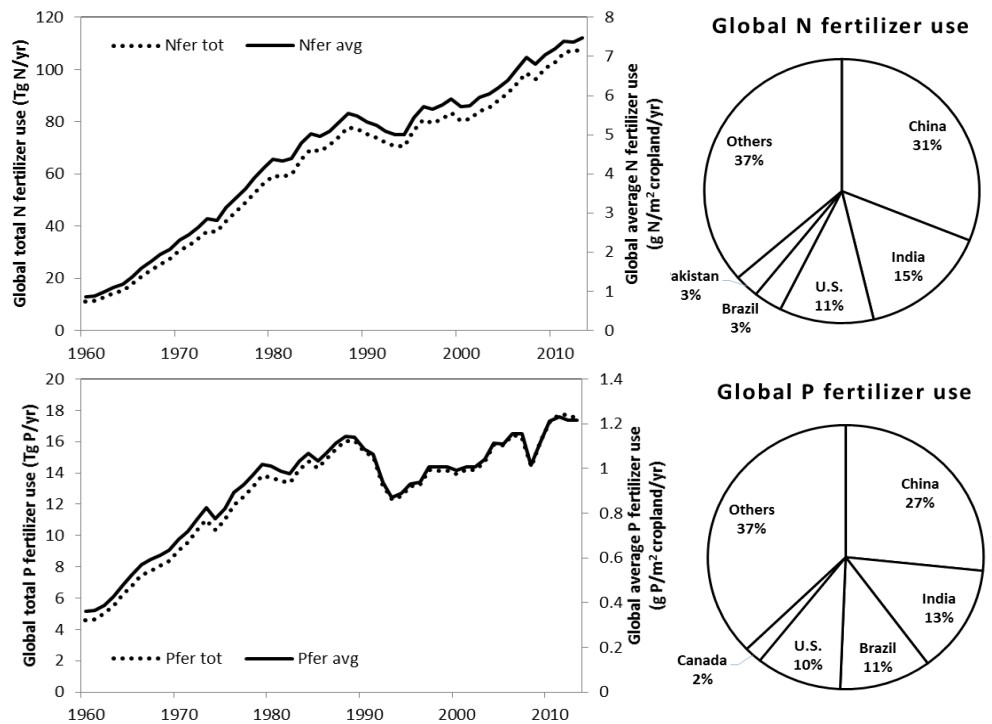


Figure 3 Temporal patterns of global nitrogen (N) and phosphorous (P) fertilizer use in terms of total amount (tot) and average rate on per-unit cropland area (avg) per year. Pie charts show the proportion of N and P fertilizer use in the top five fertilizer-consuming countries and others in the year of 2013.




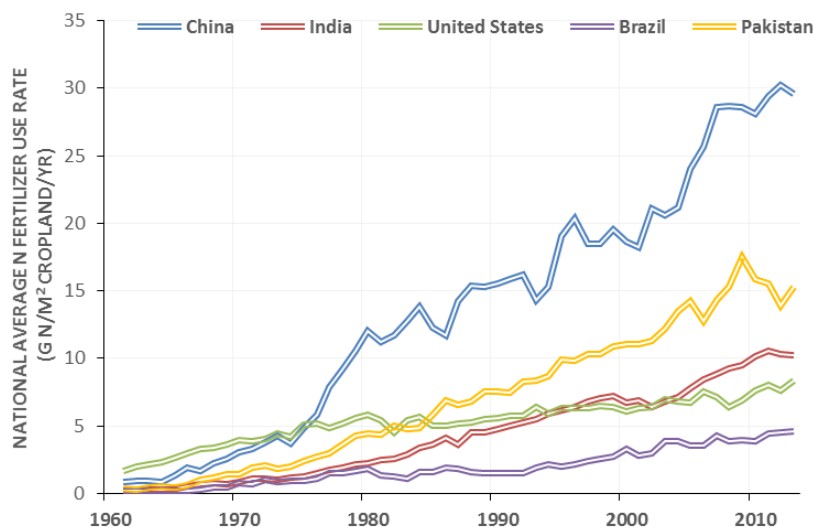


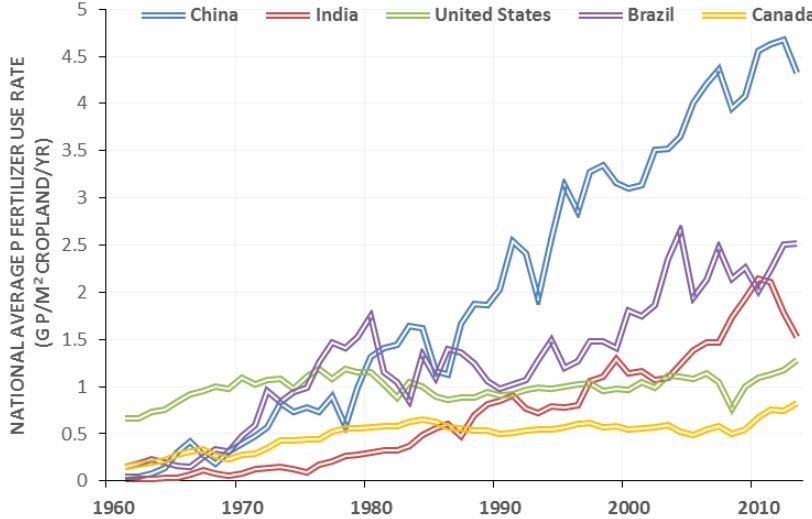


Figure 4 Interannual variations in national average N and P fertilizer use rate (g N or g P /m²
cropland/yr) in the top five fertilizer consuming countries during 1961-2013





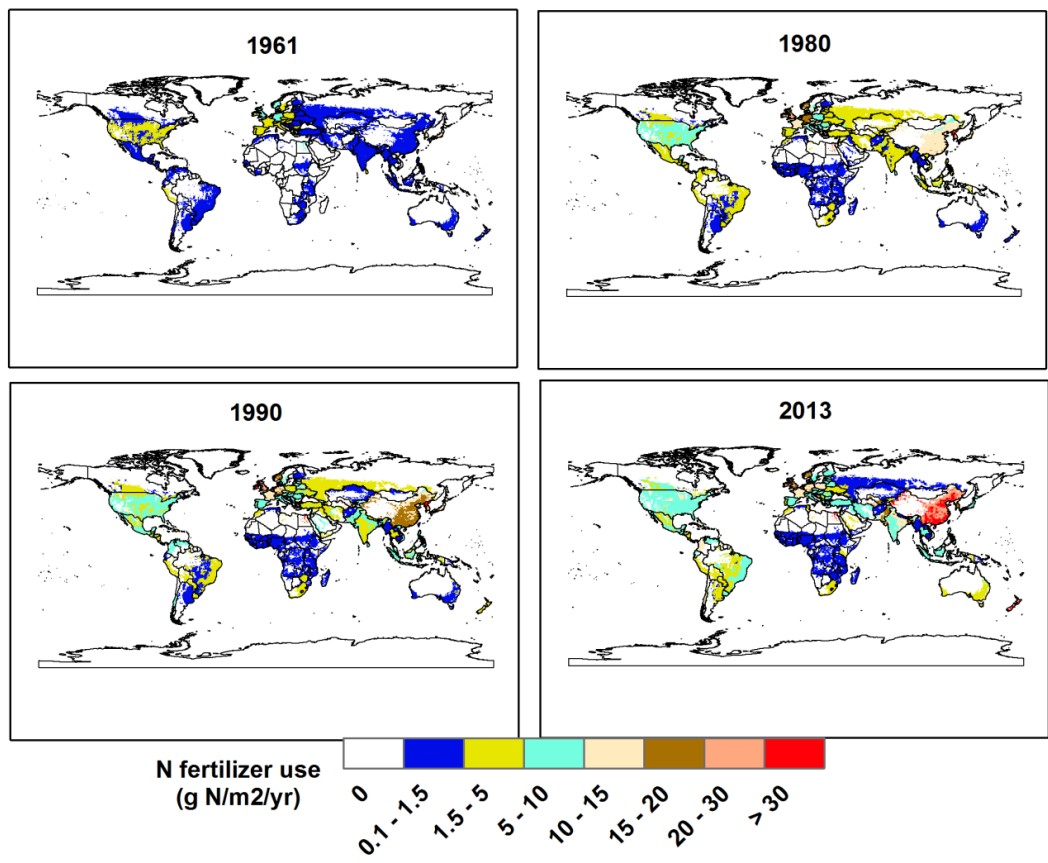


Figure 5 Spatial distribution of global agricultural nitrogen (N) fertilizer use in the year of 1961,
1980, 1990 and 2013. Colors show N fertilizer use rate in per m² cropland of each pixel.





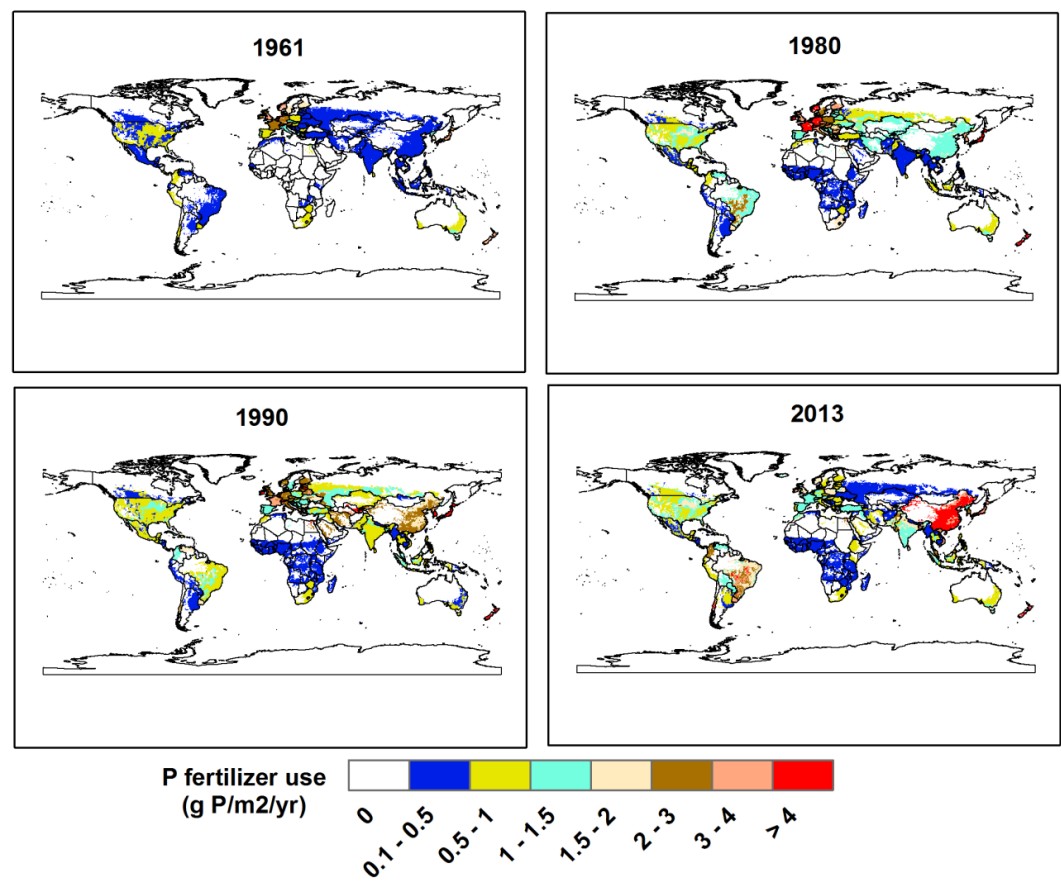


Figure 6 Spatial distribution of global agricultural phosphorus (P) fertilizer use in the year of
1961, 1980, 1990, and 2013. Colors show P fertilizer use rate in per m$^2$ cropland of each pixel.



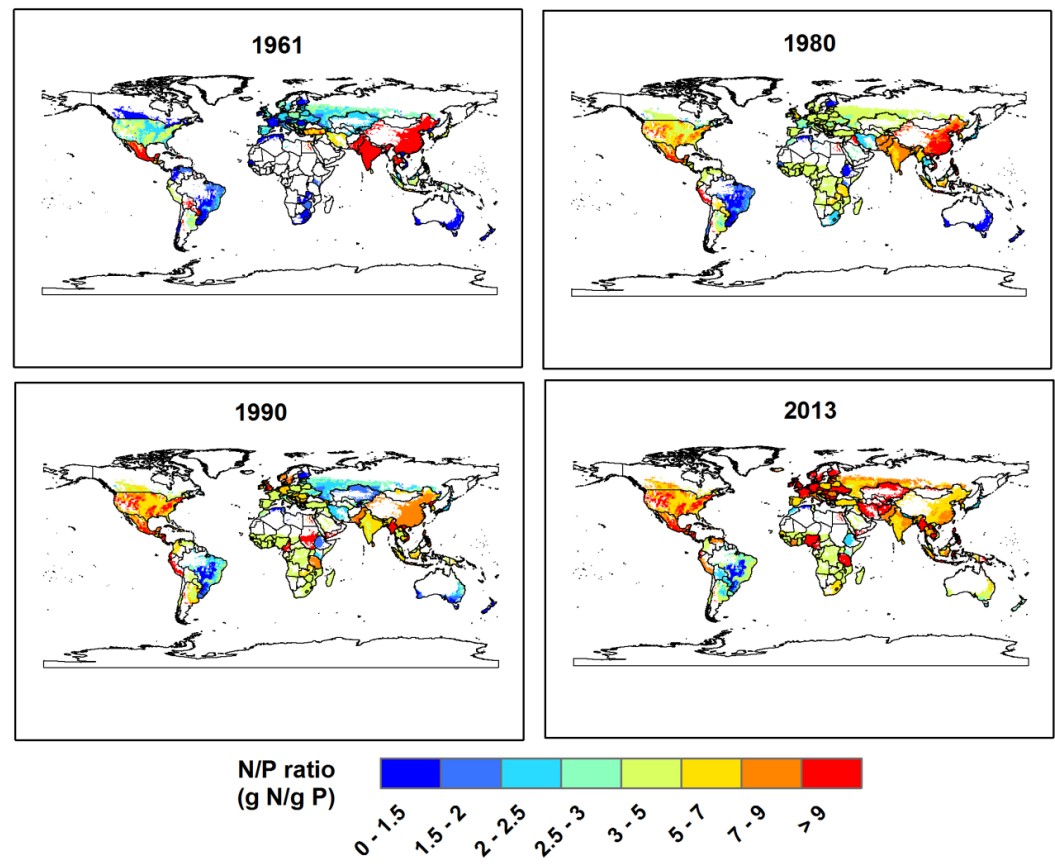


Figure 7 Spatial distribution and changes of N/P ratio in synthetic fertilizer application across the
world in the years of 1961, 1980, 1990, and 2013





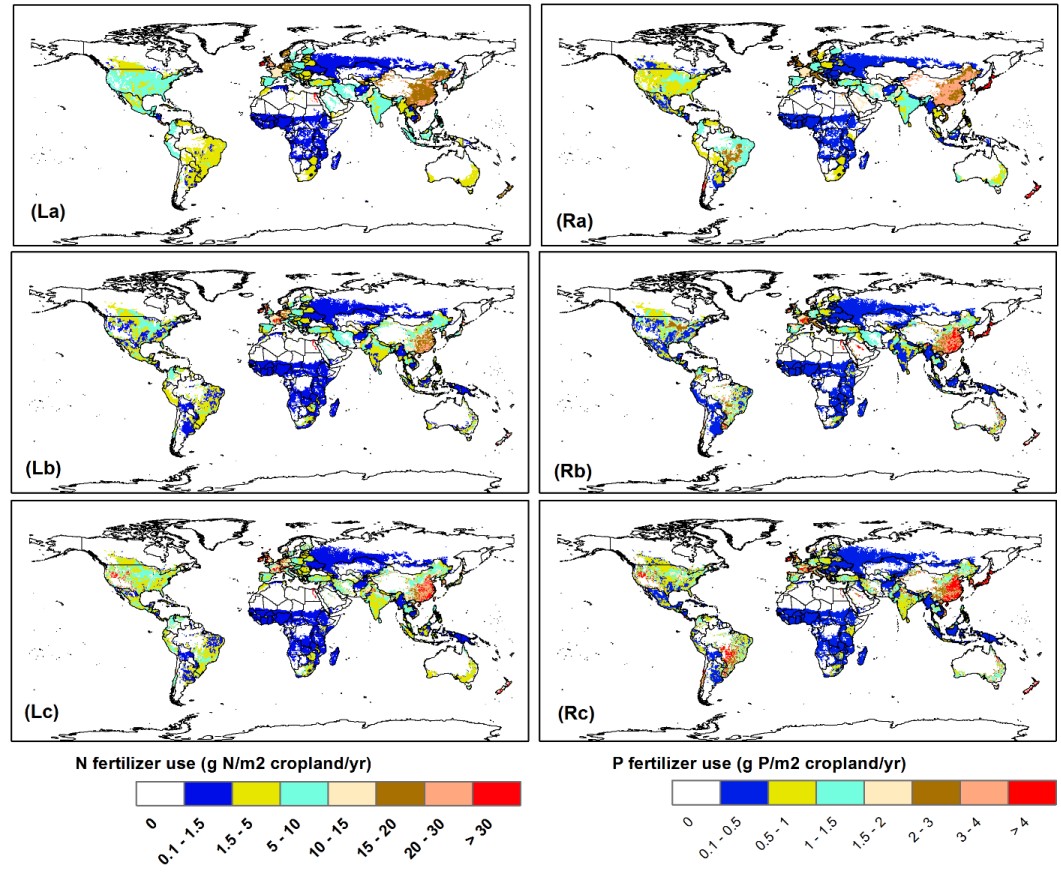


Figure 8 Comparison of global N and P fertilizer use maps from this study (panel a), Potter et al.,
2010 (panel b), and Mueller et al. 2012 (panel c) in the year 2000. Left panels (La-Lc) indicate N
fertilizer use rate and Right panels (Ra-Rc) for P fertilizer use in the unit of g N or P /m$^2$
cropland/yr.

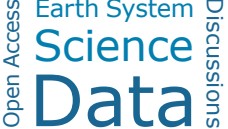

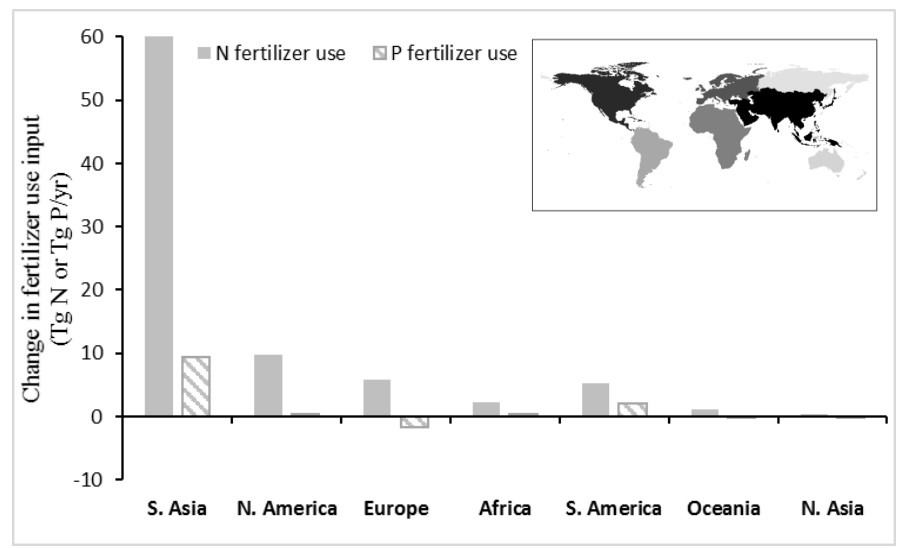


Figure 9 Changes in N and P fertilizer use (Tg N or Tg P/yr) between the 1960s and recent 5
years (2009-2013). Upper right panel shows delineation of seven continents across globe.



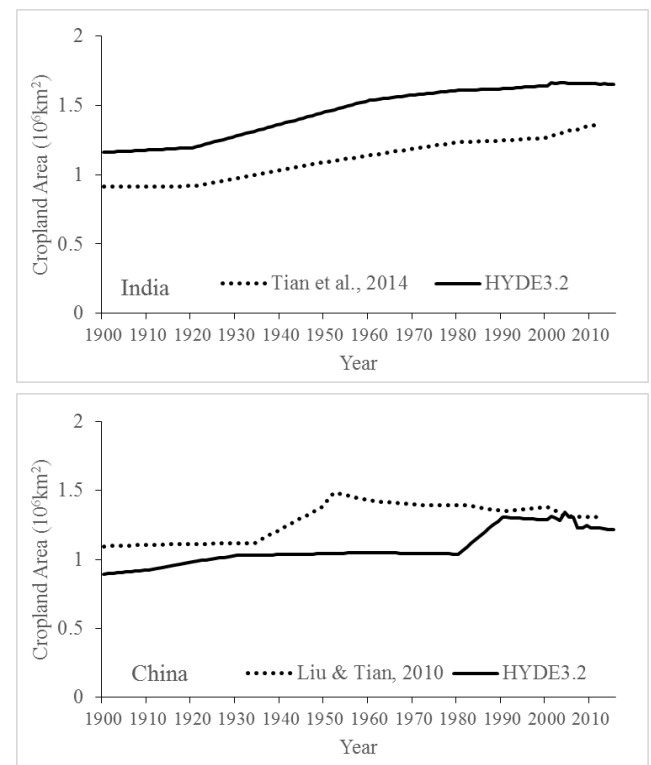


Figure 10 Differences of historical cropland area between high-resolution satellite-derived
regional LCLUC data (China: Liu and Tian, 2010; India: Tian et al., 2014) and HYDE 3.2 (Klein
Goldewijk, 2016) during 1900-2013.