# Peer review of "Global nitrogen and phosphorus fertilizer use for agriculture production in the past half century: Shifted hot spots and nutrient imbalance"

_Earth System Science Data, 2016_

## Referee Comment (RC1) · Anonymous Referee #1 · 7 Sep 2016

This is an interesting paper that describes a dataset of nitrogen and phosphorus fertilizer use over a period of several decades. This brings me immediately to my first concern about the work, that is the sentence starting on line 59 saying that there is a lack of long-term long term spatially explicit datasets for fertilizer use. There have been several publications with data starting in 1970, for example in the framework of the GLOBAL-NEWS project published in special issues of GBC in 2005 and 2009-2011, and later comparisons have been made of a number of such datasets. It is peculiar that the authors have not even referenced these studies and also a more recent one covering the full 20th century published in PNAS in 2013.

[Figure]

I also have some concerns regarding the methods used. N fertilizer use is assumed to linearly increase from zero in 1910 to the first data year in FAOSTAT (1961). This is incorrect, because the N fertilizer use in industrialized countries started to explode just after the second World War, particularly in the late 1940s and early 1950s. P fertilizer use was already important in the early 20th century. So I do not think these assumptions are realistic. Early data can be found in the first production yearbooks from FAO, I think published in 1949 when FAO was still based in Washington, DC. This data includes estimates for the 1930s.

The IFA data on fertilizer use by crop are frequently published for a limited number of countries. IFA, FAO and IFDC have published fertilizer use by crop for a much larger number of countries, I think the latest of such publications was in 2003. Although the limited number of countries represent 94% of global fertilizer use, it is useful to have data for smaller countries as well. In addition, the data now do not cover grassland and meadows, while the older datasets have all this information. So in that way grassland could have been covered in a better way than it is now (if at all). In addition, by using the whole series of IFA data, a much longer time period could have been based on real estimates for a larger number of countries instead of extrapolations.

A further concern is that harvested are is used to distribute fertilizer, but perhaps a better way could have been crop yields, which combines area and biomass production.

My major concern is that the authors have only made an inventory of synthetic fertilizers. For biogeochemical studies, or studies on emissions of greenhouse gases or the carbon cycle, also the inputs of animal manure are needed, and biological N fixation, deposition, etc. So just fertilizer use is incomplete, and to assist researcher in need of nutrient data at the global scale, such datasets would be extremely helpful. Just fertilizers is only a part of the nutrient use in agriculture.

So the authors have not reviewed the literature and did not start with the knowledge already available, both approaches and statistical information, which is a pity, because

most of this data is open access and freely available. In addition, the data as published in this discussions paper in itself is not really useful, since it lacks other sources of nutrients that in many countries are more important than synthetic fertilizers.

---

## Editor Comment (EC1) · D. J. Carlson (Editor) · 17 Sep 2016

Notes on ESSD 2016-24 and ESSD-2016-35.

Both products achieve the same global resolution (0.5 x 0.5 degrees over global land areas) for approximately similar time periods (1961 to 2010 in one case, 1961 to 2013 in the other case). Both report total synthetic N applied as chemical fertiliser. One elaborates NO3 and NH4 components of the total N, the other adds total P. One starts from country self-reported fertiliser use statistics (from FAO) while the other starts from industry reported fertiliser consumption records (IFA). Both use identical third party crop

area data (e.g. Monfreda) but different historical land use data. Both adopt the year 2000 for intercomparison and validation purposes. Both report very similar increases in total global use of N fertilisers over the time period but they differ slightly in their discussion of geographic and country-specific use patterns over time.

If, as I suspect, both data sets achieve positive reviews, e.g. seem likely to prove useful to readers and subsequent users, and presuming that from the separate review processes ESSD would not designate one or the other data set as preferred, then subsequent users will necessarily need to make a choice between somewhat similar data sets. In that case it seems fair and useful, and a proper use of the open discussion process, to pose a short series of questions to both sets of authors, and to expect that the separate responses should provide a guide to unique aspects and strengths of each data set.

How does the choice of different starting sources, FAOSTAT vs IFA, influence the subsequent processing and overall quality of the derived product?

Does the difference in tactics adopted to deal with variable completeness of country data (imputation to fill gaps in one case and focus on primarily the largest fertiliser users in the other case) induce a substantial or insubstantial difference in the outcomes of the two data production efforts.

Both sets of authors compare their products to Potter et al. 2010 and specifically for the year 2000. If each set of authors now includes the other data set in that comparison, do their overall conclusions change?

What specific information about time histories or geographic patterns of fertiliser use do readers and users gain from the inclusion of NH4 and NO3 data in the one case and from the inclusion of P data in the other case?

Finally, how does each set of authors see their efforts and products as complimentary to the other effort?

---

## Referee Comment (RC2) · Anonymous Referee #2 · 30 Sep 2016

This paper presents a global 0.5-degree resolution spatially-explicit annual time-series N and P fertilizer use data for the period of 1900-2013 by combining country-level fertilizer use record, crop-specific fertilizer use data, global maps of annual cropland area. While for the period of 1900-1960, some assumptions and simplifications were used due to the limitations in the input IFA/FAO data used for this work.

This harmonized data set is beneficial and valuable by providing a consistent set of long-term spatially-explicit time-series synthesis fertilizer use information, which can be directly used as an input driver for a variety of global- and regional-scale modeling

and assessment researches. Even though there is possibility to improve the quality of this data set by including additional fertilizer use information as inputs, e.g. early IFA/FAO/IFDC data and province/state-level fertilizer use data in select countries, this data set still makes one step forward, compared with other existing similar data sets, by providing both crop type-based spatial variation and longer time period for fertilizer use rate data.

The methodology described in this paper needs further clarification. Specifically,

1. There is no mention of which country boundary polygon layers were used in this study.

2. In section "Crop-specific N and P fertilizer use rate", it's unclear whether the spatial resolution of "grid cell g" is 0.5-degree or same as the resolution of Monfreda data (i.e. 5-arc min).

3. In section "Crop-specific N and P fertilizer use rate", there is no mention of how the situation of a grid cell sitting on the boundary of multiple adjacent countries was handled.

4. In section "Harmonizing national total and crop-specific fertilizer use rate", it's unclear the spatial resolution of "grid cell g" is 0.5-degree or same as the resolution of HYDE data (i.e. 5-arc min)

5. Figure 1 only contains data elements. It can be improved by:

1) Including process elements to form a full workflow diagram. Data and process elements can be represented in different shapes, for example, data in oval and process in rectangle;

2) Adding metadata information (e.g. spatial resolution and time period) into data elements to better describe the characteristics of each data element

6. Using "Crop-specific N and P fertilizer use rate" to calculate fertilizer use rate for all

other years suppose to be based on assumptions:

1) Relative fertilizer use rates across 13 crop groups in each country was assumed to be unchanged within the whole time frame of this study

2) Crop groups mixing ratio in each grid was assumed to be unchanged within the whole time frame of this study

Both those two assumptions could introduce uncertainties into the final data set. But in the "Uncertainty and future needs" section, only bullet 1) was mentioned, while not bullet 2).

---

## Author Comment (AC1) · 9 Nov 2016

Thanks for your precious comments. We summarized your points and responded as below.

1. Authors states the lack of long-term spatially explicit dataset for fertilizer use, but there are data starting from 1970, and recent paper covering the full 20th century published in PNAS in 2013 is not cited

Response: Many century-long earth system modeling studies require data set of agricultural nitrogen fertilizer use starting from the beginning of last century. And most

existing geospatial data of nitrogen fertilizer use are based on FAO or IFA country-level survey, but ignore the spatial heterogeneity and cross-crop variation in fertilizer application within country. Therefore, to generate a century-long geospatial data set by considering both cross-country and within-country heterogeneity is our motivation for this study. We are not sure if the paper you mentioned is Bouwman et al. (2013, PNAS) or not. If so, this research only provides estimates of synthetic nitrogen fertilizer use in a few historical time points (e.g., 1900, 1950, 2000). We quote the approach they used for generating these estimates, "Fertilizer use for 1950 is taken from the FAO (13). For the year 1900, we used country data on the use of fixed N (industrially produced N fertilizer, Chili nitrate, guano, coke-oven ammonium sulfate, calcium cyanamide, and electric-arc calcium nitrate) for 1913 (48) and assumed that the use in each country in 1900 is 80% of that in 1913. For the year 2000, we used country data on total synthetic fertilizer consumption and crop production and animal stocks (22) and N and P fertilizer use by crop (49)." It cannot meet the data requirement of an ecosystem/earth system modeling study. We cited this paper in the revised manuscript. The similar data development can be found in Bouwman et al. (2005) for the time points of 1970, 1995, and 2030.

2. The assumption of linear increase of N and P fertilizer use prior to 1961 is problematic.

Response: Thanks for raising this question. Since the data before 1961 is hard to found and has much divergence in its quality and reporting criteria among countries, we removed the description of this part from the manuscript and only focus on the period 1961-2013.

3. Older version of IFA data (e.g., 2003) covers more countries and fertilizer use in grassland and meadows.

Response: Thanks for your suggestion. The IFA data of fertilizer use by crop (Heffer, 2013) we are using now depicts crop-specific fertilizer use in the year of 2010-2010/11.

[Figure]

It is the most up-to-date country-level crop-specific fertilizer use data we can get so far. It covers over 94% of global fertilizer consumption. For the rest ∼6% fertilizer consumptions, in order to keep the data record consistent from the same data source, we used IFA-derived country-level data by assuming fertilizer use rate is uniform for all the crop types in each of these less-intensive fertilizer use countries. Since the purpose of this study is to provide long-term synthetic nitrogen fertilizer use data across global cropland, the land use data we used here only include cropland distribution and change (e.g., M3-crop, HYDE 3.2 data). We clarified this in the revised manuscript.

4. A further concern is that harvested are is used to distribute fertilizer, but perhaps a better way could have been crop yields, which combines area and biomass production.

Response: Thanks for your suggestion. We do see data development approaches of future fertilizer use scenarios that includes crop yield or biomass production (e.g., Bouwman et al., 2005, 2013), but this method ignores the changes in crop use efficiency of nitrogen and phosphorous. To facilitate modeling assessment of crop yield and agricultural production, we would like to have a set of fertilizer use data that is independent of productivity to avoid autocorrelation and problematic assumptions. Therefore, we only use harvested cropland area to spatialize country-level fertilizer use amount.

5. Synthetic fertilizer use is only part of nutrient inputs in agriculture and incomplete for biogeochemical studies

Response: We agree that there are many sources of nutrient input, such as atmospheric nitrogen deposition, biological nitrogen fixation, manure N and P application, weathering, that have important influence to biogeochemical cycling. However, the purpose of this study is to develop spatially-explicit time-series data of agricultural N and P fertilizer use across the globe, rather than to estimate all the nutrient input sources.

6. So the authors have not reviewed the literature and did not start with the knowledge already available, both approaches and statistical information, which is a pity, because

most of this data is open access and freely available. In addition, the data as published in this discussions paper in itself is not really useful, since it lacks other sources of nutrients that in many countries are more important than synthetic fertilizers.

Response: We have compared the newly-developed data in this study with other two typical similar data sets that demonstrated gridded maps of agricultural fertilizer uses (Potter et al., 2010; Mueller et al., 2012). The other data sets failed to provide similar information, either due to limited spatial extent or temporal span, or the lack of geospatial component. As we mentioned above, there are indeed many sources of nutrient input, among which synthetic fertilizer use accounts for 1/3 and half of total N and P inputs, respectively. It's of critical importance to know the spatial and temporal patterns of N and P fertilizer uses across global cropland. It could help us identify the hot spots and hot periods of anthropogenic nutrient input, and the consequent changes in food production, environmental quality, and climate change, and lead to effective nutrient management toward a sustainable world.

Reference

Bouwman, A.F., Van Drecht, G., Knoop, J.M., Beusen, A.H.W. and Meinardi, C.R., 2005. Exploring changes in river nitrogen export to the world's oceans. Global Biogeochemical Cycles, 19(1). DOI: 10.1029/2004GB002314 Bouwman, L., Goldewijk, K.K., Van Der Hoek, K.W., Beusen, A.H., Van Vuuren, D.P., Willems, J., Rufino, M.C. and Stehfest, E., 2013. Exploring global changes in nitrogen and phosphorus cycles in agriculture induced by livestock production over the 1900–2050 period. Proceedings of the National Academy of Sciences, 110(52), pp.20882-20887. Mueller, N.D., Gerber, J.S., Johnston, M., Ray, D.K., Ramankutty, N. and Foley, J.A., 2012. Closing yield gaps through nutrient and water management. Nature, 490(7419), pp.254-257. Potter, P., Ramankutty, N., Bennett, E.M. and Donner, S.D., 2010. Characterizing the spatial patterns of global fertilizer application and manure production. Earth Interactions, 14(2), pp.1-22.

---

## Author Comment (AC2) · 9 Nov 2016

Notes on ESSD 2016-24 and ESSD-2016-35.

1. Both products achieve the same global resolution (0.5 x 0.5 degrees over global land areas) for approximately similar time periods (1961 to 2010 in one case, 1961 to 2013 in the other case). Both report total synthetic N applied as chemical fertiliser. One elaborates NO3 and NH4 components of the total N, the other adds total P. One starts from country self-reported fertilizer use statistics (from FAO) while the other starts from industry reported fertiliser consumption records (IFA). Both use identical third party crop

area data (e.g. Monfreda) but different historical land use data. Both adopt the year 2000 for intercomparison and validation purposes. Both report very similar increases in total global use of N fertilisers over the time period but they differ slightly in their discussion of geographic and country-specific use patterns over time. If, as I suspect, both data sets achieve positive reviews, e.g. seem likely to prove useful to readers and subsequent users, and presuming that from the separate review processes ESSD would not designate one or the other data set as preferred, then subsequent users will necessarily need to make a choice between somewhat similar data sets. In that case it seems fair and useful, and a proper use of the open discussion process, to pose a short series of questions to both sets of authors, and to expect that the separate responses should provide a guide to unique aspects and strengths of each data set.

Response: Thanks for your careful reading and positive comments. There are some differences between these two data products that may influence the choice of future users. Although we both used crop area data from Monfreda et al (2008, M3-crops data), it's for different purpose. In our study, combined with IFA crop-specific fertilizer use data, the harvested area and crop type distribution revealed from M3-crops data have been used to calculate the area-weighted crop fertilizer use rate in each grid cell, and to allocate national fertilizer use amount based on crop distribution. Therefore, cross-crop divergence in using fertilizer has been considered and displayed in our data. However, ESSD 2016-24 used M3-crops data to identify dominant crop type, and combining crop calendar data (Sacks et al., 2010) to decide the timing of fertilizer use in each grid cell. We have addressed your questions and more difference aspects between this study and the other data product (ESSD 2016-24 hereafter) as below.

2. How does the choice of different starting sources, FAOSTAT vs IFA, influence the subsequent processing and overall quality of the derived product?

Response: In this study, we adopted the data of fertilizer use by crop from IFA (Heffer, 2013) to spatialize country-level fertilizer use amount derived from IFA into time-series gridded maps. Therefore, this data product demonstrates both cross-country

and within-country heterogeneity in fertilizer uses by considering different fertilizer use level among crop types (e.g., N fertilizer use rates are quite different in corn and soybean). We use crop-specific data and country-level fertilizer survey both from IFA to make the estimate consistent.

3. Does the difference in tactics adopted to deal with variable completeness of country data (imputation to fill gaps in one case and focus on primarily the largest fertilizer users in the other case) induce a substantial or insubstantial difference in the outcomes of the two data production efforts.

Response: Thanks for pointing out this. We would like to clarify that our study doesn't only focus on the largest fertilizer use, but overlooked the rest. We used IFA data of fertilizer use by crop that covers over 94% of global fertilizer consumption, to spatialize country-level fertilizer use amount onto grid cells. But for the rest less intensive fertilizer use countries (consuming ∼6% of global fertilizer), we adopted IFA time-series country and continental record (removing the reported countries) of fertilizer use rate, by assuming uniform fertilizer use among crop types in each country. We clarified this in our revised manuscript and redrew figure 2 to demonstrate countries with and without crop-specific fertilizer use in IFA data and those countries excluded by IFA.

Overall, this approach captured spatial cross-crop variation in fertilizer use rate across those intensive fertilizer using countries, while keeping the global total consistent with IFA data record. We think the substantial difference between these two data sets is not from the resulted global total fertilizer use amount we reported or the way we fill gaps (i.e. use continental average rate with reported countries removed in this study, or use covariate information in ESSD 2016-24), but from the spatial heterogeneity we presented in these two data sets (i.e., cross-crop within-country heterogeneity in fertilizer use in this study, and country-level uniform fertilizer use rate in ESSD 2016-24).

4. Both sets of authors compare their products to Potter et al. 2010 and specifically for the year 2000. If each set of authors now includes the other data set in that comparison,

do their overall conclusions change?

Response: In addition to Potter et al. 2010, we compared our data with Mueller et al. (2012) that also demonstrate N and P fertilizer use rate at gridded level (Figure 8). Our approach is more like the way that Potter et al. (2010) used in their study by considering weighted crop-specific fertilizer use to spatialize fertilizer pattern within a country. If compared in global fertilizer consumption amount, our study is very close to IFA and FAO record (e.g., N fertilizer use in 2000 is estimated as 82.1 Tg N/yr by IFA, 80.8 Tg N/yr by FAOSTAT, and ours is 80.1 Tg N/yr, Table 1), while ESSD 2016-24 has a higher estimate of 85 Tg N/yr in the same year. Compared to ESSD 2016-24, our data provides more spatial details based on distribution of crop types (through M3-crop), IFA data of fertilizer use by crop in the most intensive fertilizer use countries, and IFA country- and continental fertilizer use data in the rest countries, while they used national consumption data from FAOSTAT and statistical gap filling to depict spatial pattern of fertilizer use.

5. What specific information about time histories or geographic patterns of fertiliser use do readers and users gain from the inclusion of NH4 and NO3 data in the one case and from the inclusion of P data in the other case?

Response: In our case, we didn't split N fertilizer into NH4 and NO3, but include P fertilizer data since it provides more information on anthropogenic nutrient input and has important implications on historical stoichiometric changes. We anticipate this data could better serve the Earth System Modeling communities by demonstrating cross-country, cross-crop variations in fertilizer uses (both N and P), and revealing the shift of hot spots of nutrient input across the globe during 1961 to 2013. It can inform both field and modeling studies of the spatial and temporal changes in agricultural fertilizer uses, and assist complex assessment and decision support in effective balanced nutrient (both N and P) management in the future.

6. Finally, how does each set of authors see their efforts and products as complimentary to the other effort?

Response: Thanks for this great question. We summarized a few things in our study that could compliment ESSD 2016-24 as below: 1) As described above, our study demonstrates both cross-country and cross-crop heterogeneity in fertilizer uses by compiling a number of data sources (e.g., annual record of national fertilizer use amount, crop-specific fertilizer use, crop type and its gridded distribution across the globe, time-series land use data). It covers global cropland areas with consideration of their annual change during the period 1961-2013. 2) Our study includes both agricultural N and P fertilizer use rate, which is important anthropogenic nutrient source that could substantially contribute to global food production, biogeochemical cycles, greenhouse gas balance, climate change, and riverine nutrient export from land to coastal oceans. Ratio of agricultural N and P fertilizer input and its change could give us potential explanation to global nutrient imbalance and ecosystem stoichiometric trend. 3) With finer-scale spatial variation in this study, we reveal the hot spots of agricultural fertilizer use shifted from the U.S. and western Europe to East Asia for N, and from Europe to Central China and small area of Brazil for P over the past half century. It is helpful in understanding the spatial shift of environmental consequences of agricultural nutrient enrichment and forecasting the future trend of earth system responses.

Reference

Heffer, P., 2013. Assessment of fertilizer use by crop at the global level 2010-2010/11. International Fertilizer Industry Association, Paris, http://www.fertilizer.org/ItemDetail?iProductCode=9592Pdf&Category=STAT&WebsiteKey
Monfreda, C., Ramankutty, N. and Foley, J.A., 2008. Farming the planet: 2. Geographic distribution of crop areas, yields, physiological types, and net primary production in the year 2000. Global biogeochemical cycles, 22(1). DOI: 10.1029/2007GB002947
Mueller, N.D., Gerber, J.S., Johnston, M., Ray, D.K., Ramankutty, N. and Foley, J.A., 2012. Closing yield gaps through nutrient and water management. Nature, 490(7419), pp.254-257. Potter, P., Ramankutty, N., Bennett, E.M. and Donner, S.D.,
2010. Characterizing the spatial patterns of global fertilizer application and manure production. Earth Interactions, 14(2), pp.1-22. Sacks, W. J., Deryng, D., Foley, J. A., and Ramankutty, N.: Crop planting dates: an analysis of global patterns, Global Ecology and Biogeography, 19, 607–620, 2010.

---

## Author Comment (AC3) · 9 Nov 2016

Thanks a lot for your positive comments. We have revised the manuscript according to your advice. Please find the detailed revision as below:

1. There is no mention of which country boundary polygon layers were used in this study.

Response: we added the data source (GADM database of Global Administrative Areas) and its citation in manuscript.

2. In section "Crop-specific N and P fertilizer use rate", it's unclear whether the spatial resolution of "grid cell g" is 0.5-degree or same as the resolution of Monfreda data (i.e. 5-arc min).

Response: it's 0.5-degree. We have clarified it in text.

3. In section "Crop-specific N and P fertilizer use rate", there is no mention of how the situation of a grid cell sitting on the boundary of multiple adjacent countries was handled.

Response: We converted shapefile of country boundary to half-degree grid file, and then used it to calculate harvested areas of 13 crop groups in each country based on M3-crop map aggregated to the same resolution. There is no grid cell "sitting on the boundary of multiple adjacent country".

4. In section "Harmonizing national total and crop-specific fertilizer use rate", it's unclear the spatial resolution of "grid cell g" is 0.5-degree or same as the resolution of HYDE data (i.e. 5-arc min)

Response: It is 0.5 degree. We have clarified this in the text.

5. Figure 1 only contains data elements. It can be improved by: 1) Including process elements to form a full workflow diagram. Data and process elements can be represented in different shapes, for example, data in oval and process in rectangle; 2) Adding metadata information (e.g. spatial resolution and time period) into data elements to better describe the characteristics of each data element

Response: Thanks for your constructive suggestions. We re-drew figure 1 by including process elements and adding more metadata information for specific data sources. Details can be found the new figure.

6. Using "Crop-specific N and P fertilizer use rate" to calculate fertilizer use rate for all other years suppose to be based on assumptions: 1) Relative fertilizer use rates across 13 crop groups in each country was assumed to be unchanged within the whole

time frame of this study 2) Crop groups mixing ratio in each grid was assumed to be unchanged within the whole time frame of this study Both those two assumptions could introduce uncertainties into the final data set. But in the "Uncertainty and future needs" section, only bullet 1) was mentioned, while not bullet 2).

Response: Thank you for pointing this out. We extended the text to address the possible uncertainty introduced by the second assumption, and potential data improvement given the future availability of time-series crop distribution maps.
* * *